# Sodium-Glucose Co-Transporter 2 Inhibitors Reduce Macular Edema in Patients with *Diabetes mellitus*

**DOI:** 10.3390/life12050692

**Published:** 2022-05-06

**Authors:** Tomoaki Tatsumi, Toshiyuki Oshitari, Yoko Takatsuna, Ryoichi Ishibashi, Masaya Koshizaka, Yuki Shiko, Takayuki Baba, Koutaro Yokote, Shuichi Yamamoto

**Affiliations:** 1Department of Ophthalmology and Visual Science, Chiba University Graduate School of Medicine, 1-8-1, Inohana, Chuo-ku, Chiba 260-8670, Chiba, Japan; tarii@aol.com (T.O.); yoko_takatsuna@chibah.johas.go.jp (Y.T.); babatakayuki@nifty.com (T.B.); shuyama@faculty.chiba-u.jp (S.Y.); 2Department of Ophthalmology, International University of Health and Welfare School of Medicine, 4-3, Kozunomori, Narita 286-8686, Chiba, Japan; 3Department of Ophthalmology, Chiba Rosai Hospital, 2-16, Tatsumidaihigashi, Ichihara 290-0003, Chiba, Japan; 4Department of Endocrinology, Hematology, and Gerontology, Chiba University Graduate School of Medicine, 1-8-1, Inohana, Chuo-ku, Chiba 260-8670, Chiba, Japan; bassyna6ce1484@yahoo.co.jp (R.I.); overslope@chiba-u.jp (M.K.); koutaroyokote@gmail.com (K.Y.); 5Department of Medicine, Division of Diabetes, Endocrinology and Metabolism, Kimitsu Chuo Hospital, Kisarazu 292-0822, Chiba, Japan; 6Biostatistics Section, Clinical Research Center, Chiba University Hospital, 1-8-1, Inohana, Chuo-ku, Chiba 260-8670, Chiba, Japan; yuki11290219@gmail.com

**Keywords:** aflibercept, anti-VEGF, diabetic macular edema, sodium-glucose co-transporter 2 inhibitors, SGLT2

## Abstract

Purpose: To determine the efficacy of systemic sodium-glucose co-transporter 2 inhibitors (SGLT2i) on diabetic macular edema (DME). Methods: The medical records of patients with DME with a central retinal thickness (CRT) ≥320 µm in men and 305 µm in women, more than 6 months after the initiation of diabetes mellitus treatment, were reviewed. The CRT and best-corrected visual acuity (BCVA) were evaluated before and after the initiation of systemic SGLT2i and non-SGLT2i treatments. Results: There were 24 eyes of 19 patients with DME that were treatment naïve or had not received treatments for the DME within four months before the initiation of SGLT2i. In these patients, the BCVA had a 0.31 ± 0.39 logarithm of the minimum angle of resolution (logMAR) units at the baseline, and it did not improve significantly at 0.26 ± 0.29 logMAR units after the initiation of SGLT2i (*p* = 0.56). However, the SGLT2i treatment significantly reduced the CRT from 423.3 ± 79.8 µm to 379.6 ± 69.5 µm (*p* = 0.0001). In the same evaluation of 19 eyes of 14 patients with DME that were initiated with non-SGLT2i agents, there was no significant difference between the baseline BCVA and the BCVA after the initiation of non-SGLT2i (*p* = 0.47). The CRT increased significantly after the initiation of non-SGLT2i (*p* = 0.0011). In three eyes in which the SGLT2i treatments were administered at the time of anti-vascular endothelial growth factor (VEGF) treatments, the anti-VEGF treatment alone had only a limited effect on the DME, but the reduction in the DME was enhanced after the addition of SGLT2i. Conclusions: These findings indicate that systemic SGLT2i can reduce DMEs, and they suggest that SGLT2i may be an additional treatment option to anti-VEGF treatments for eyes with DMEs.

## 1. Introduction

Diabetic macular edema (DME) is the most common retinal abnormality that can lead to a reduction in the vision of patients with non-proliferative diabetic retinopathy (NPDR) [1]. A meta-analysis has shown that 6.81% of 22,896 individuals with diabetes had DME [2]. Intravitreal injections of anti-vascular endothelial growth factor (anti-VEGF) agents have become the global standard treatment for DME [3,4]. However, there is a high rate of recurrence, and frequent injections are required to maintain visual acuity [5]. In a different study, it was reported that DME persisted in 31.6–65.6% of patients even after multiple anti-VEGF injections [6]. The treatment of DME refractory to anti-VEGF treatment is a challenging problem [7,8].

Systemic sodium-glucose co-transporter 2 inhibitors (SGLT2i) have become one of the standard treatments for type 2 diabetes mellitus. The mechanism for the effect of SGLT2i is their ability to inhibit glucose reabsorption in the proximal tubules and allowing glucose to be excreted in the urine. This then improves the glycemic levels of glucose [9]. In addition, SGLT2i has a diuretic effect. Large-scale clinical trials have shown that SGLT2i-treated patients had a lower risk of cardiovascular death and hospitalization for heart failure [10,11]. In addition, systemic SGLT2i treatments were associated with a slower progression of kidney disease and lower rates of clinically relevant renal events [12].

Laboratory studies have shown that SGLT2 is present in bovine retinal pericytes and rat mesangial cells [13,14]. Hyperglycemia results in the loss of pericytes which leads to pathological, functional and morphological changes, and these pathological changes can be attenuated by the application of SGLT2i [14]. Although SGLT2i have been shown to improve the general condition of patients with diabetes, the effects of SGLT2i on eyes with DME have not been determined.

There have been reports of DME cases in which macular edema improved after the initiation of SGLT2i [15,16,17], and there exists a hypothesis that SGLT2i improves macular edemas in DME cases. However, this hypothesis has not been tested.

Thus, the purpose of this study was to determine the efficacy of systemic SGLT2i in resolving DME. 

## 2. Methods

The medical records of patients with DME who were examined by ophthalmologists at the Chiba University Hospital from August 2014 to February 2022 and whose internal medications for type 2 diabetes mellitus could be confirmed were reviewed. We determined the effects of systemic SGLT2i on DMEs by comparing the best-corrected visual acuity (BCVA) and central retinal thickness (CRT) before and after the initiation of systemic SGLT2i. Patients who had not undergone any DME treatments such as anti-VEGF agents, steroids, and retinal photocoagulation at the time of the initiation of SGLT2i treatment were classified as SGLT2i monotherapy patients and the effects of SGLT2i were evaluated. The data obtained immediately before the beginning of treatment with SGLT2i were used as the baseline data. Patients whose data were not known for the six weeks before the initiation of SGLT2i were excluded. The data obtained from 2 to 15 weeks after the initiation of SGLT2i were used as the data for the after-initiation of SGLT2i data. All of these data were used for the statistical analyses, but in the graphs, the data closest to 8 weeks after the initiation of SGLT2i were used as the representative data and plotted. If other treatments for DME such as anti-VEGF treatments were administered after SGLT2i was initiated, the data before the treatment was used to exclude the effects of the other treatment. Patients who had received treatments such as anti-VEGF treatment, steroids, retinal photocoagulation, cataract surgery and vitrectomy within 6 months before the initiation of SGLT2i were excluded. Patients whose initial SGLT2i treatment began within 6 months of the start of diabetes treatment were also excluded to eliminate the effects of rapid changes in the systemic condition.

To evaluate the effects of SGLT2i in eyes with DME, patients whose start time of SGLT2i was specified, with a CRT ≥ 320 µm for men and 305 µm for women [18] either before or after the initiation of SGLT2i, were studied. These patients who added SGLT2i or changed to SGLT2i diabetes treatment were included in the SGLT2i group. Patients who added non-SGLT2i or changed to non-SGLT2i diabetes treatment were placed in the control group (non-SGLT2i group), and changes in their DMEs were evaluated before and after the initiation of non-SGLT2i treatment. The exclusion criteria such as ophthalmic treatment (anti-VEGF, photocoagulation, steroid treatment, and others) and start time of diabetes treatment that may affect DME were the same as in the SGLT2i group.

Patients treated with anti-VEGF agents before and after the initiation of SGLT2i were evaluated separately from SGLT2i monotherapy patients because it was difficult to evaluate the efficacy of SGLT2i. The effects of anti-VEGF treatments on DMEs before and after the initiation of SGLT2i were studied by examining the clinical timelines. These patients were classified as SGLT2i and anti-VEGF combination therapy patients and the effects of SGLT2i were evaluated.

Eyes with an epiretinal membrane, vitreomacular traction syndrome, active uveitis, other retinal diseases, and severe glaucoma were excluded. In addition, patients with prior cerebrovascular disorders such as cerebral infarction and myocardial infarction were also excluded. Sustained-release steroid agents were not used because they were not approved in Japan.

The BCVA was measured with a Landolt chart and with the same protocol for all patients. The CRT was measured in the images obtained by spectral domain optical coherence tomography (SD-OCT; Heidelberg Engineering, Heidelberg, Germany). If the criteria for recurrence of macular edema were met, anti-VEGF treatments were performed.

All patients underwent an intravitreal injection of aflibercept (IVA) using a standard method without complications. IVA was performed with a regimen that consisted of an intravitreal injection at the time of diagnosis and re-administration at the recurrence of the DME, i.e., the 1 + PRN or 3 + PRN regimen. The patients were examined every four weeks. For the IVA, 2 mg of aflibercept was injected intravitreally.

### 2.1. Ethical Statements

This study was approved by the Institutional Review Board of the Chiba University Graduates School of Medicine (IRB#3866 approved 11 September 2020). Written informed consent was obtained from patients treated with DME. However, this study was a retrospective study, and if it was difficult to obtain a patient’s written informed consent, consent was obtained in the form of opt-out. Informed consent in this case is replaced by guaranteeing the opportunity for patients to decline the use of their data such as their medical records in the study. Matters related to this study were posted on the website of the Department of Ophthalmology. If one of the subjects of this study confirmed the intention of refusal, the data of the subject were excluded from the study.

### 2.2. Statistical Analyses

Statistical analyses were performed using Microsoft Excel and JMP pro 16 (SAS Institute, Cary, NC, USA). The data are expressed as the mean ± standard deviations (SDs) for numerical variables and as a numerical percentage for categorical variables. The baseline characteristics of the patients in the two groups were compared using *t*-tests for numerical variables and Chi-square tests or Fisher’s exact tests for categorical variables.

The decimal visual acuities were converted to the logarithm of the minimum angle of resolution (logMAR) units for the statistical analyses. The significance of the differences in the findings was determined with Student’s *t*-tests or a linear mixed-effects model. Student’s *t*-tests were used to compare numerical data between two groups. When analyzing changes in CRT and BCVA before and after the initiation of SGLT2 inhibitors, both eyes contributed to the analysis in five cases, and the data were measured and analyzed 1 to 3 times for each eye after initiation. It was assumed that both eyes in the same case were correlated. Therefore, a linear mixed-effects model was upheld. For the mixed model, we designated the cases and eyes (case) as a random effect (random intercept), while the period (week) after initiation of SGLT2i or non-SGLT2i was taken to be the fixed effect. All tests were two-tailed, and *p* < 0.05 was considered statistically significant.

## 3. Results

### 3.1. Number of Patients and Baseline Characteristics of Patients in SGLT2i Group and Non-SGLT2i Group

There were 24 eyes of 19 patients with DME in the SGLT2i group and 19 eyes of 14 patients included in the non-SGLT2i group. The baseline characteristics of the patients in both groups are shown in Table 1. The diabetes treatment drugs for the 19 eyes of the non-SGLT2i group were sulfonylurea (SU) in five eyes, dipeptidyl peptidase-4 inhibitor (DPP-4i) in five eyes, glucagon-like peptide-1 receptor agonist (GLP-1 RA) in four eyes, metformin in two eyes, thiazolidinediones (TZD) in two eyes, and a combination of α-glucosidase inhibitor (α-GI) and glinide in one eye (Appendix A).

### 3.2. Effects of Systemic SGLT2i Monotherapy

Clinical course of each case in the SGLT2i group is shown in Table 2. In the SGLT2i group, the mean baseline CRT before the SGLT2i treatment was 423.3 ± 79.8 µm. The data closest to the 8th week after the initiation of SGLT2i were used as the representative data. The CRT after initiation of SGLT2i was 379.6 ± 69.5 µm (Figure 1a), and it reduced significantly after the initiation of SGLT2i treatment (slope = −4.49, 95% CI, −6.62 to −2.36; *p* = 0.0001, mixed-effects model using all data obtained).

The baseline BCVA was 0.314 ± 0.389 logMAR units (Snellen 20/41) and the BCVA after initiation of SGLT2i (representative data) was 0.259 ± 0.289 logMAR units (Snellen 20/36; Figure 1b). The BCVA did not improve significantly after the initiation of SGLT2i (slope = −0.0023, 95% CI, −0.010 to 0.0056; *p* = 0.56, mixed-effects model using all data obtained).

Clinical course of each case in the non-SGLT2i group is shown in Table 3. In the non-SGLT2i group, the mean baseline CRT before the non-SGLT2i treatment was 374.3 ± 79.4 µm, and the CRT after initiation of the non-SGLT2i agents (representative data) was 387.6 ± 92.3 µm (Figure 1c). The increase in the CRT was significant after the non-SGLT2i treatment (slope = 3.07, 95% CI, 1.31 to 4.82; *p =* 0.0011, mixed-effects model using all data obtained).

The baseline BCVA was 0.139 ± 0.241 logMAR units (Snellen 20/28), and the BCVA after initiation of the non-SGLT2i treatment was 0.177 ± 0.269 logMAR units (Snellen 20/30) (Figure 1d). The BCVA (logMAR) did not improve significantly after the initiation of the non-SGLT2i treatment (slope = 0.0017, 95% CI, −0.0031 to 0.0066; *p* = 0.47, mixed-effects model using all data obtained).

A scatter plot of the CRT and BCVA for the 24 eyes before and after the initiation of SGLT2i is shown in Figure 2a,b. Four of the twenty-four eyes demonstrated a decrease in the CRT of more than 20%, and one eye had an increase in the CRT of more than 20% after the initiation of SGLT2i. One of the 24 eyes had an improvement of the BCVA of 0.2 logMAR units, and one eye demonstrated a decrease in the BCVA. Scatter plots of the CRT and BCVA for the 19 eyes before and after the initiation of non-SGLT2i treatment are shown in Figure 2c,d.

The patterns of changes in the DMEs [19] are shown in Appendix A for each patient. Sixteen eyes had sponge-like retinal swelling and eight eyes had cystoid macular edema (CME). The mean baseline CRT was 423.3 ± 92.3 µm in patients with the sponge-like pattern and 423.5 ± 45.5 µm in patients with CME (*p* = 0.99). The mean CRT after initiation of SGLT2i was reduced to 380.6 ± 76.2 µm and 377.6 ± 53.6 µm, respectively. The CRT was reduced significantly in both types (slope = −5.31, 95% CI, −8.74 to −1.88, *p* = 0.0039, and slope = −3.20, 95% CI, −5.10 to −1.30; *p* = 0.0029, respectively, mixed-effects model). However, the BCVA did not improve significantly in both types (slope = −0.0040, 95% CI, −0.017 to 0.0087; *p* = 0.52, and slope = 0.00085, 95% CI, −0.0070 to 0.0087; *p* = 0.82, respectively, mixed-effects model).

### 3.3. Effects of SGLT2i in Combination with Anti-VEGF Agents

Overall, 20 eyes of 13 patients were treated with anti-VEGF agents before and after the initiation of SGLT2i. It was difficult to evaluate the effect of SGLT2i on DME in 17 of 20 eyes due to the effect of anti-VEGF treatment. However, the DME improved after the initiation of SGLT2i in three eyes that were refractory to anti-VEGF treatments. The clinical course of these three eyes was as follows.

Case 1. A 66-year-old woman had been diagnosed with proliferative diabetic retinopathy (PDR) and both eyes underwent pan-retinal photocoagulation and cataract surgery. She was also treated with a sub-Tenon injection of triamcinolone acetonide (STTA) twice but the effects were limited. Six months after the second STTA injection, IVA was started with the 3 + PRN regimen. At the initiation of the IVA, the CRT of her left eye was 858 µm, and the BCVA was 0.82 logMAR units (Snellen 20/133). After three IVAs and one subthreshold laser application, the CRT was 797 µm with a lack of improvement of the DME. Therefore, another IVA was scheduled four weeks later, but an examination prior to the injection showed that the CRT had decreased to 437 µm. At this time, it was found that 5 mg of oral dapagliflozin, a commercial SGLT2i, had been started by her general physician three weeks earlier in place of linagliptin, a dipeptidyl peptidase-4 inhibitor. After that, the ME decreased further, and the CRT decreased to 212 µm and the BCVA to 0.52 logMAR units (Snellen 20/67). The course of the CRT in this case is shown in Figure 3.

Case 2. A 56-year-old man had been diagnosed with PDR in his left eye 1.5 years prior. Pan-retinal photocoagulation was performed but the CRT increased to 720 μm 12 months later. Therefore, IVA was initiated under the 1 + PRN regimen. Just before the IVA, the CRT of his left eye was 558 µm and the BCVA was 0.52 logMAR units (Snellen 20/67). After four anti-VEGF treatments, the ME improved slightly and the CRT was 491 µm and the BCVA was 0.82 logMAR units (Snellen 20/133). At about the same time, his internist started 2.5 mg luseogliflozin treatment, an SGLT2i treatment, and 500 mg metformin. Four weeks after the IVA, the BCVA was 0.70 logMAR units (Snellen 20/100) and the CRT had improved to 245 µm. The DME recurred 8 weeks later but the CRT was 372 µm, which was smaller than that before the initiation of SGLT2i. Two more IVA treatments improved the CRT. After that, no recurrence of the ME was observed even without IVA treatment for more than 48 weeks. The course of the CRT in this case is plotted in Figure 4.

Case 3. A 71-year-old man was diagnosed with NPDR with DME. IVA was begun under the 3 + PRN regimen. At the initiation of the IVA, the CRT of his right eye was 558 µm and the BCVA was 0.52 logMAR units (Snellen 20/67). The DME did not improve even after five IVAs and the CRT remained at 576 μm. After that, systemic SGLT2i was initiated, and 2 weeks later, the CRT decreased to 476 µm. With additional SGLT2i, the CRT decreased to 236 µm, and the BCVA improved to 0.046 logMAR units (Snellen 20/22). The course of the CRT in this case is plotted in Figure 5.

## 4. Discussion

There have been three studies conducted on the effectiveness of SGLT2i for the treatment of DME [15,16,17]. One study on 10 vitrectomized eyes of five patients showed that the CRT improved from 500.5 μm to 410 μm and the BCVA improved from 0.35 to 0.15 logMAR units at 3 months after the initiation of SGLT2i [15]. Another study reported on a DME patient who changed treatment from metformin hydrochloride (250 mg/day) to ipragliflozin (25 mg/day), which is an SGLT2i. This case was not treated with anti-VEGF agents, but steroid treatment before the initiation of SGLT2i had almost no effect. After the switch to SGLT2i, the CRT improved from 762 μm to 589 μm [16]. The other study reported on three patients with chronic DME who experienced a notable improvement of the DME after the initiation of SGLT2i. In these three patients, the DMEs improved significantly in both eyes after the SGLT2i treatment without any other ophthalmic treatment [17].

In our cohort, systemic SGLT2i treatment also reduced the DMEs significantly. On the other hand, when a non-SGLT2i treatment was applied, the DME increased. It is believed that if these cases had not been treated for DME, the DMEs would have worsened. The SGLT2i group is believed to have the same background, and the DME was nevertheless improved. However, an improvement of the CRT by 20% or more was achieved in only 4 of 24 eyes (16.7%). Thus, SGLT2i may be a treatment option for DME, but it may be difficult to obtain good therapeutic effects for all DME patients with this treatment alone.

It is generally believed that better therapeutic effects can be obtained by combining SGLT2i with anti-VEGF treatments. Twenty-four eyes of nineteen patients underwent SGLT2i monotherapy, and there were five cases in which both eyes were the subject of this study. In these five patients, if the DMEs in both eyes showed the same response to SGLT2i, this could affect the statistical findings. However, previous studies have reported that patients who had been treated with systemic SGLT2i did not always have the same response and course in both eyes [15,17]. Therefore, we included patients in whom both eyes were treated.

In this analysis, we designated the periods (week) after initiation of SGLT2i or non-SGLT2i as a fixed effect. We were not comparing the baseline and the data after the initiation of SGLT2i or non-SGLT2i but analyzing changes over time using measured data 1 to 3 times after the initiation. In the SGLT2i group, the CRT also significantly reduced when comparing the baseline with the representative data after the initiation of SGLT2i (*p* = 0.0003, *t*-test). However, in the non-SGLT2i group, there was no significant difference in the CRT between the baseline and the representative data after the initiation of non-SGLT2i (*p* = 0.18, *t*-test). It is believed that this was not because non-SGLT2i significantly increased CRT, but because DME was active and tended to worsen.

Cases 1 and 2 were refractory to anti-VEGF treatment, but the DME decreased rapidly after the initiation of systemic SGLT2i treatment. In these cases, the reduction of the DME was attributed to SGLT2i. However, it was difficult to evaluate the combined effects of SGLT2i and anti-VEGF agents accurately because we cannot eliminate the possibility that the effects of the previous anti-VEGF treatment may have had an influence.

It has been reported that SGLT2i treatments may be effective in reducing DME [15,16,17], but the effect of SGLT2i on DME has not been definitively determined. In addition, its mechanism of action has not been established. It is generally believed that the rapid effects of SGLT2i on DME are related to their diuretic effects, and these diuretic effects may be involved in Cases 1 and 2.

In Case 3, the serum creatinine level was elevated to ≥3 mg/dL, and renal dysfunction occurred. Usually, SGLT2i is not used to treat diabetes mellitus in patients with renal dysfunction because it cannot be expected to have a hypoglycemic effect. SGLT2i was used in these cases to prevent kidney disease. It is believed that the macular edema may have been improved due to the continuation of anti-VEGF treatments regardless of SGLT2i. However, if SGLT2i was effective in improving DME, it was probably not due to its diuretic effect.

It has been reported that SGLT2 is present in retinal pericytes and mesangial cells, and it has been reported that SGLT2 inhibitors may protect diabetic retinopathy through direct actions on the retinal pericytes [14]. In Case 3, this direct action on the retinal pericytes, rather than its diuretic effect, may have been the cause of the gradual improvement of the macular edema.

It is clear that anti-VEGF treatment is effective for DME but due to the financial limitations, a combined therapy of anti-VEGF treatments with pan-retinal photocoagulation, focal photocoagulations, or local injections of triamcinolone acetonide (TA) is preferred. In addition, most ophthalmologists use anti-VEGF treatment as the first-line therapy but prefer the 1 + *pro re nata* (PRN) regimen [20]. In Japan, the mean number of anti-VEGF injections is about three/year, and ophthalmologists have tried to improve the therapeutic effects fewer anti-VEGF injections [21,22]. Retinal experts in Japan have suggested that it is important to establish a multipronged treatment strategy for DME [23]. These multimodal treatments have centered on the use of anti-VEGF treatments with steroid therapy [7,8], laser photocoagulation, or vitrectomy [24].

Earlier studies have shown that SGLT2i were effective in treating DME [15,16,17], and our study showed that the CRT in patients with DME was significantly reduced after the initiation of SGLT2i. Thus, SGLT2i is a treatment option for DME.

DME is caused by a number of complex factors, [23] and the release of VEGF is the leading factor. However, there are other factors that need to be considered. Although the mechanism of action for SGLT2i has not been determined definitively, it is believed that the mechanism is different from that of anti-VEGF agents. Cases refractory to anti-VEGF agents are relatively strongly affected by factors other than VEGF. It is believed that the SGLT2i were effective in Cases 1 and 2. SGLT2i may be more effective against DME when used in combination with anti-VEGF agents.

The combination of albuminuria and diabetic retinopathy (DR) portends a higher cardiovascular risk [25]. Microangiopathy, expressed as albuminuria, is correlated with cardiovascular (CV) risk and can identify patients at high CV risk. In this way, multifactorial intensive therapy can be of great benefit when mitigating the risk of major fatal and non-fatal cardiovascular events and the mortality in high-risk diabetic kidney disease patients [26]. DR/DME is associated with other modes of expression of microangiopathy, and an accurate diagnosis of these is important not only to reduce the risk of progression to vision loss, but also to identify patients at high CV risk.

The diagnosis of DR/DME is often delayed. Recently, the development and deployment of digital technology has progressed, and the importance of telemedicine is increasing. Telemedicine has been shown to be effective and useful for the screening for DR and DME, and a follow-up of DR and DME is very difficult without this method, such as during the COVID-19 pandemic [27,28]. Telemedicine brings specialists of DME treatment closer to clinical diabetes centers. If SGLT2i becomes a treatment option for DME, it may be compatible and useful with telemedicine during pandemics.

There are limitations in this study. This was a retrospective study with its inherent biases. Another limitation was the small number of patients. Considering that SGLT2i are drugs that according to the American Diabetes Association (ADA) and the European Association for the Study of Diabetes (EASD) guidelines [29,30] can be used from the early stage of diabetes management, it is possible to study its effects in a larger number of cases. Therefore, the results of this study are preliminary, and it is necessary to study such a treatment in a larger number of cases. Other limitations include the lack of standard protocol for the treatments, and lack of systemic data and information on the other complications of diabetes. The effects of a patient’s general condition were not sufficiently evaluated and analyzed. In this study, we report our findings in three eyes in which SGLT2i was considered to be effective in a combination therapy with SGLT2i and anti-VEGF treatment. However, in these cases, the paradigm of therapeutic agents varied, and it was difficult to evaluate which agent was effective. There was no control group for anti-VEGF treatment alone, and it is difficult to evaluate whether its combination with SGLT2i was effective or if anti-VEGF alone was effective.

When SGLT2i was used in combination with anti-VEGF agents, it was difficult to evaluate the effect of SGLT2i because anti-VEGF agents are highly effective for DME. Thus, further large prospective clinical trials are needed to determine the efficacy of SGLT2i for DME. We are conducting a multicenter, randomized, open-label trial to evaluate the safety and efficacy of SGLT2i in resolving DME with a combination of anti-VEGF agents and SGLT2i in patients with type 2 diabetes and DME [31].

## 5. Conclusions

In conclusion, the results showed that SGLT2i can reduce DMEs in patients with DR. However, not all patients with DMEs can be adequately treated with SGLT2i treatment alone, and it is ineffective in some patients. We presented three cases in which the DME improved after beginning systemic SGLT2i treatment in eyes that were refractory to anti-VEGF treatments. We suggest that SGLT2i may be an additional treatment option to anti-VEGF treatment.

## Figures and Tables

**Figure 1 life-12-00692-f001:**
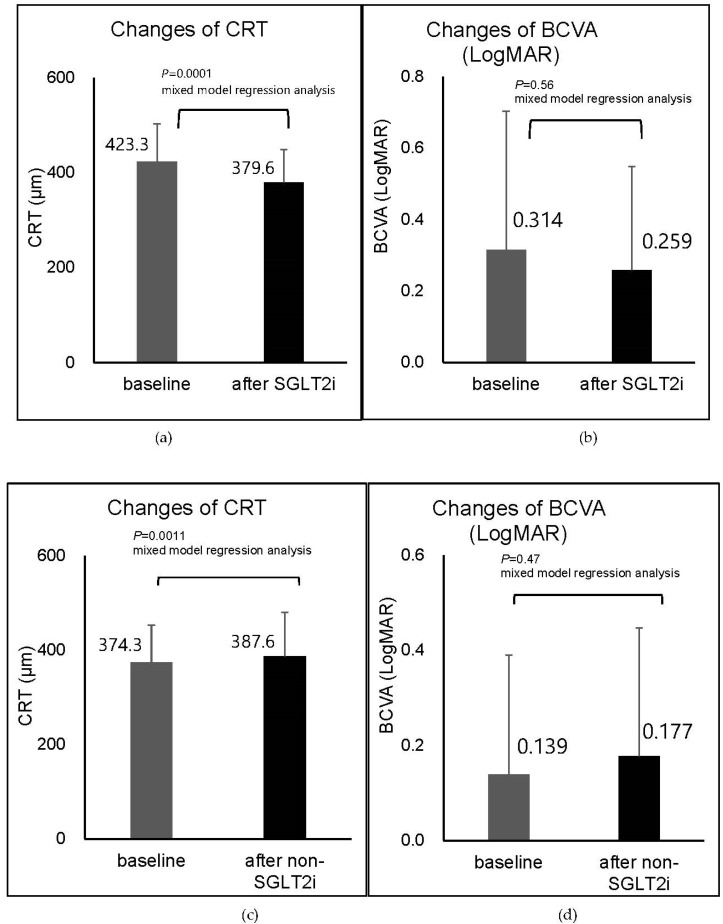
Comparisons of the central retinal thickness (CRT). (**a**) Best-corrected visual acuity (BCVA) in logarithm of the minimum angle of resolution (logMAR) units (**b**) before and after the initiation of sodium-glucose co-transporter 2 inhibitors (SGLT2i) in 24 eyes. Comparisons of CRT (**c**), BCVA (logMAR) (**d**) before and after the initiation of non-SGLT2i in 19 eyes. The baseline CRT increased significantly after initiation of non-SGLT2i (*p* = 0.0011), whereas CRT significantly reduced after initiation of SGLT2i (*p* = 0.0001). There were no significant changes in the BCVAs in either group. The data closest to the 8th week after the initiation of SGLT2i or non-SGLT2i were used as the representative data.

**Figure 2 life-12-00692-f002:**
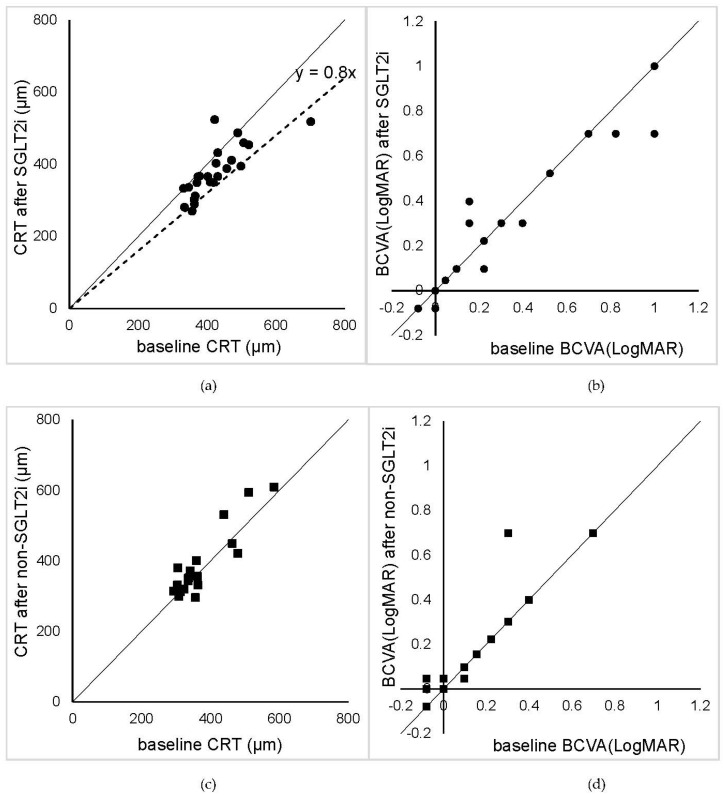
Scatter plots of central retinal thickness (CRT) (**a**) and best-corrected visual acuity (BCVA) in logarithm of the minimum angle of resolution (logMAR) units (**b**) before and after the beginning of sodium-glucose co-transporter 2 inhibitors (SGLT2i) treatment in 24 eyes. Scatter plots of CRT (**c**) and BCVA (logMAR) (**d**) before and after the beginning of non-SGLT2i in 19 eyes. The solid line represents y = x, and the dashed line in (**a**) represents y = 0.8x. The plots below the dashed line represent cases where the CRT decreased by 20% or more. Overall, 4 of the 24 eyes experienced a decrease in the CRT of more than 20%, and one eye experienced an increase in the CRT after the initiation of SGLT2i. The data closest to the 8th week after the initiation of SGLT2i or non-SGLT2i was taken as the representative data.

**Figure 3 life-12-00692-f003:**
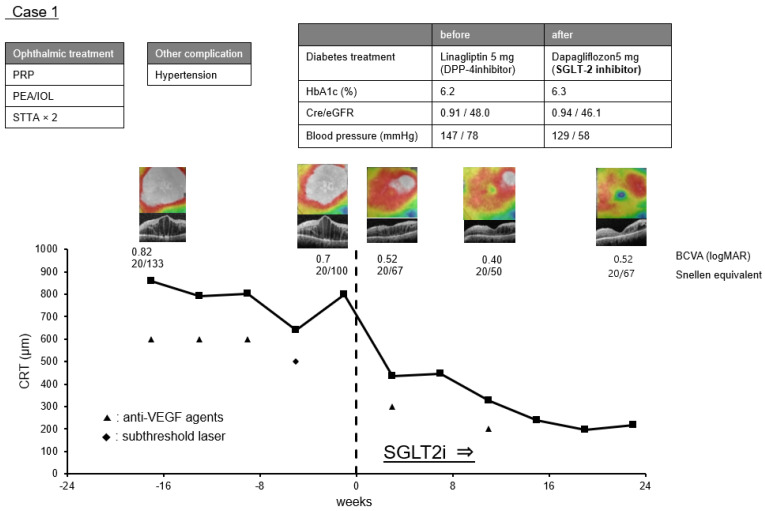
Clinical course before and after the initiation of sodium-glucose co-transporter 2 inhibitors (SGLT2i) in Case 1. The graph shows the time course of central retinal thickness (CRT), optical coherence tomography images, and best-corrected visual acuity (BCVA) in logarithm of the minimum angle of resolution (logMAR) units and Snellen equivalent at each time point. The history of ophthalmic treatment, other complication of diabetes mellitus and therapeutic agents for diabetes before and after the initiation of SGLT2i, HbA1c, serum creatinine (cre), estimated glomerular filtration rate (eGFR), and systemic blood pressure are shown above.

**Figure 4 life-12-00692-f004:**
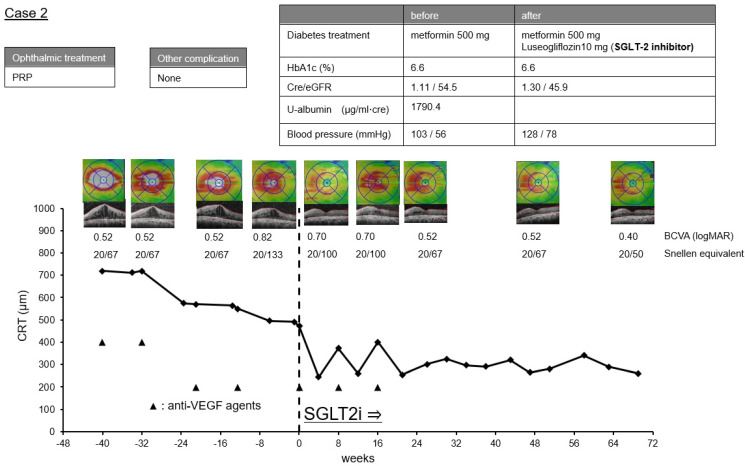
Clinical course before and after the initiation of sodium-glucose co-transporter 2 inhibitors (SGLT2i) in Case 2. The graph shows the time course of central retinal thickness (CRT), optical coherence tomographic images, and the BCVA in logMAR units and Snellen equivalent at each time point. The history of ophthalmic treatment, other complication of diabetes mellitus and therapeutic agents for diabetes before and after the initiation of SGLT2i, HbA1c, serum creatinine (cre), estimated glomerular filtration rate (eGFR) and systemic blood pressure are shown above.

**Figure 5 life-12-00692-f005:**
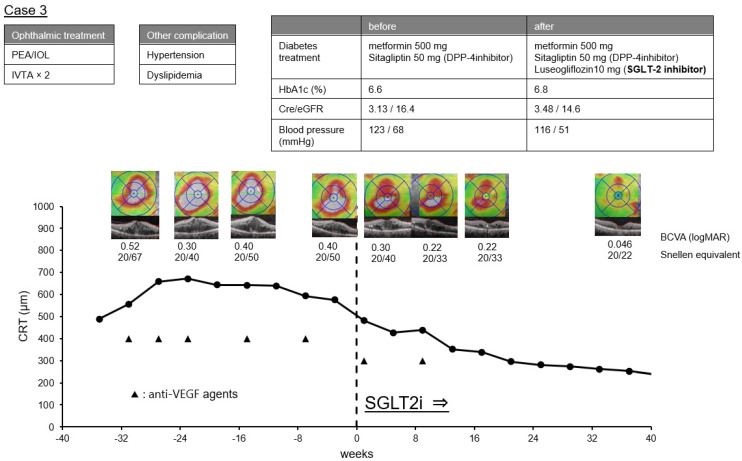
Clinical course before and after the initiation of sodium-glucose co-transporter 2 inhibitors (SGLT2i) in Case 3. The graph shows the time course of central retinal thickness (CRT), optical coherence tomographic images, and the BCVA in logMAR units and Snellen equivalent at each time point. The history of ophthalmic treatment, other complication of diabetes mellitus and therapeutic agents for diabetes before and after the initiation of SGLT2i, HbA1c, serum creatinine (cre), estimated glomerular filtration rate (eGFR) and systemic blood pressure are shown above.

**Table 1 life-12-00692-t001:** The baseline characteristics of patients.

		SGLT2i Group (19 Patients 24 Eyes)		Non-SGLT2i Group (14 Patients 19 Eyes)		*p*-Value
Sex	Male, *n* (%)	11	(57.9)	7	(53.8)	Fisher’s exact test
	Female, *n* (%)	8	(42.1)	7	(46.2)	*p* = 0.73
Age, years	Mean (S.D)	61.6	(9.7)	61.6	(7.8)	*t*-test
						*p* = 0.997
	<50, *n* (%)	3	(15.8)	0	(0)	Chi-square test
	50≤ <60, *n* (%)	4	(21.1)	5	(35.7)	*p* = 0.33
	60≤ <70, *n* (%)	6	(31.6)	6	(42.9)	
	70≤ <80, *n* (%)	6	(31.6))	3	(21.4)	
	80≤, *n* (%)	0	(0)	0	(0)	
Subject eye	Right, *n* (%)	10	(41.7)	12	(63.2)	Fisher’s exact test
	Left, *n* (%)	14	(58.3)	7	(36.8)	*p* = 0.22
Mean interval between DME diagnosis and SGLT2i or non-SGLT2i, months	Mean (S.D)	14.3	(12.4)	9.4	(7.9)	*t*-test
						*p* = 0.21
	≤1, *n* (%)	2	(10.5)	2	(14.3)	Chi-square test
	1< ≤7, *n* (%)	4	(21.1)	5	(35.7)	*p* = 0.69
	7< ≤20, *n* (%)	9	(47.4)	4	(28.6)	
	20<, *n* (%)	4	(21.1)	3	(21.4)	
HbA1c, %	Mean (S.D)	7.18	(1.23)	7.64	(1.67)	*t*-test
						*p* = 0.38
	<6.5	4	(10.5)	2	(14.3)	Chi-square test
	6.5≤ <7.2	9	(47.4)	6	(42.9)	*p* = 0.56
	7.2≤ <7.7	3	(15.8)	1	(7.1)	
	7.7≤	3	(15.8)	5	(35.7)	
Hypertension	(+), *n* (%)	11	(57.9)	4	(28.6)	Fisher’s exact test
	(−), *n* (%)	8	(42.1)	10	(71.4)	*p* = 0.16
Dyslipidemia	(+), *n* (%)	5	(26.3)	6	(42.9)	Fisher’s exact test
	(−), *n* (%)	14	(73.7)	8	(57.1)	*p* = 0.46
cre	Mean (S.D)	0.79	(0.21)	0.80	(0.31)	*t*-test
						*p* = 0.94
eGFR, mL/min/1.73 m^2^	Mean (S.D)	77.0	(19.8)	72.5	(18.3)	*t*-test
						*p* = 0.52
Prior photocoagulation, *n* (%)	PRP	12	(50.0)	5	(26.3)	Chi-square test *p* = 0.13
	Focal PC	3	(12.5)	1	(5.3)	
	None	9	(37.5)	13	(68.4)	
Pattern of DME, *n* (%)	Sponge-like	16	(66.7)	12	(63.2)	Chi-square test *p* = 0.42
	CME	8	(33.3)	6	(31.6)	
	SRD	0	(0)	1	(5.3)	
Baseline CRT, μm	Mean (S.D)	423.3	(79.8)	374.3	(79.4)	*t*-test
						*p* = 0.057
Baseline BCVA (logMAR)	Mean (S.D)	0.314	(0.389)	0.139	(0.247)	*t*-test
						*p* = 0.10
Baseline IOP, mmHg	Mean (S.D)	14.3	(2.8)	12.5	(3.4)	*t*-test
						*p* = 0.044

SGLT2i: sodium-glucose co-transporter 2 inhibitors; SD: standard deviation; eGFR: estimated glomerular filtration rate; CRT: central retinal thickness; BCVA: best-corrected visual acuity; logMAR: logarithm of minimum angle of resolution; IOP: intraocular pressure; PRP: panretinal photocoagulation; PC: photocoagulation; VEGF: vascular endothelial growth factor; TA: triamcinolone acetonide injection; MAPC: microaneurysm photocoagulation.

**Table 2 life-12-00692-t002:** Clinical course of each case in SGLT2i group.

Case	Eye R/L	Agents Added or Changed to	CRT (µm) Baseline	CRT (µm)/Week after Initiation						BCVA (logMAR Unit) Baseline	BCVA (logMAR)/Week after Initiation					
1	R	Ipragliflozin	701	518	/10	427	/15			0.699	0.699	/10	0.699	/15		
	L	Ipragliflozin	431	432	/10	436	/15			0.398	0.301	/10	0.523	/15		
2	R	Luseogliflozin	365	313	/4	312	/10	332	/15	0.301	0.398	/4	0.301	/10	0.222	/15
	L	Luseogliflozin	362	318	/4	302	/10	307	/15	0.222	0.222	/4	0.222	/10	0.097	/15
3	R	Ipragliflozin	457	388	/4					1	0.699	/4				
	L	Ipragliflozin	498	395	/4					1.398	0.398	/4				
4	L	Empagliflozin	363	290	/2					0.222	0.097	/2				
5	R	Empagliflozin	471	411	/9					0	0	/9				
6	R	Dapagliflozin	521	480	/6	454	/8			0.824	0.699	/6	0.699	/8		
7	L	Dapagliflozin	356	270	/12					0.097	0.097	/12				
8	R	Ipragliflozin	418	385	/6	350	/12			0.155	0.398	/6	0.398	/12		
	L	Ipragliflozin	489	477	/6	487	/12			1	1.097	/6	1	/12		
9	L	Canagliflozin	431	448	/4	366	/11			0.523	0.697	/4	0.523	/11		
10	R	Empagliflozin	378	367	/2					0.046	0.046	/2				
	L	Empagliflozin	426	403	/2					0.301	0.301	/2				
11	R	Luseogliflozin	370	357	/4	349	/7			0	0	/4	−0.079	/7		
12	L	canagliflozin	334	302	/3	281	/15			0.097	0.097	/3	0.097	/15		
13	L	Luseogliflozin	402	366	/2					0	0	/2				
14	R	Luseogliflozin	331	333	/8					−0.079	−0.079	/8				
15	L	Luseogliflozin	422	524	/4					0.155	0.301	/4				
16	L	Luseogliflozin	409	351	/8					0.046	0.046	/8				
17	L	Luseogliflozin	506	460	/8					0.301	0.301	/8				
18	R	Empagliflozin	373	365	/6	376	/15			−0.079	−0.079	/6	−0.079	/15		
19	L	Empagliflozin	346	336	/5					−0.079	−0.079	/5				

SGLT2i = sodium-glucose co-transporter 2 inhibitors; CRT = central retinal thickness; BCVA = best corrected visual acuity; logMAR = logarithm of minimum angle of resolution.

**Table 3 life-12-00692-t003:** Clinical course of each case in non-SGLT2i group.

Case	Eye R/L	Agents Added or Changed to	CRT (µm) Baseline	CRT (µm)/Week after Initiation						BCVA (logMAR Unit) Baseline	BCVA (logMAR)/Week after Initiation					
1	R	SU	335	347	/4	350	/8	355	/12	0.155	0.155	/4	0.155	/8	0.097	/12
2	R	SU	313	306	/4	312	/8	307	/12	−0.079	−0.079	/4	−0.079	/8	−0.079	/12
3	L	SU	585	610	/4					0.699	0.699	/4				
4	R	SU	323	320	/4					−0.079	0	/4				
5	R	SU	304	331	/4					0	0	/4				
6	R	GLP-1 RA	511	525	/4	594	/8			0.301	0.301	/4	0.699	/8		
7	R	DPP-4i	479	420	/4	422	/8	462	/12	0.097	0.155	/4	0.097	/8	0.222	/12
8	R	DPP-4i	337	352	/8	346	/12			0.699	0.699	/8	0.398	/12		
	L	DPP-4i	463	449	/8	537	/12			−0.079	0.046	/8	−0.079	/12		
9	R	metformin	359	401	/8	388	/12			0.398	0.398	/8	0.398	/12		
	L	metformin	439	531	/8	498	/12			0	0	/8	0	/12		
10	L	GLP-1 RA	335	322	/5	343	/10	423	/14	−0.079	−0.097	/5	−0.079	/10	−0.097	/14
11	R	DPP-4i	305	380	/6	447	/11			−0.079	0.046	/6	0.097	/11		
	L	DPP-4i	341	372	/6	395	/11			0.097	0.046	/6	0	/11		
12	R	αGI + glinide	363	346	/4	331	/11			−0.079	−0.079	/4	−0.079	/11		
13	R	TZD	293	314	/4	315	/12			0	0.046	/4	0.097	/12		
	L	TZD	362	356	/4	400	/12			0.155	0.155	/4	0.222	/12		
14	R	GLP-1 RA	356	296	/5					0.222	0.222	/5				
	L	GLP-1 RA	308	300	/5					0.301	0.301	/5				

SGLT2i = sodium-glucose co-transporter 2 inhibitors; CRT = central retinal thickness; BCVA = best corrected visual acuity; logMAR = logarithm of minimum angle of resolution; SU = Sulfonylurea; GLP-1 RA = Glucagon-like peptide-1 receptor agonist; DPP-4i = Dipeptidyl peptidase-4 inhibitor; αGI =α-glucosidase inhibitor; TZD = thiazolidinediones.

## Data Availability

The data used to support the findings of this study are available upon request from the corresponding author.

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
