# Peer review of "Sodium-Glucose Co-Transporter 2 Inhibitors Reduce Macular Edema in Patients with Diabetes mellitus"

_life, 2022, doi:10.3390/life12050692_

Round 1

Reviewer 1 Report

In this study, the authors have addressed the comments that were raised in the last review, mainly about small sample size, treatment variability and lack of controls; and have mentioned these as shortcomings of the article in the Discussion section. While the impact of the results could have been greater with a larger sample size, the current study is still significant as case studies of the 3 patients where combination of SGLT2i and anti-VEGF treatments were given. No further changes are required and the study can be accepted for publication as it is.

Reviewer 2 Report

No further comments.

Reviewer 3 Report

We notice major improvements in this version of the manuscript. However, 2 isues still remain:

1. AntiVEGF use (IVA) should be clariffied: line 254 says 20 eyes with antiVEGF, however 424 says only 3 eyes received antiVEGF???

2. Is it possible to draw conclusions regarding the differences between the 2 groups? (SGLT2i and non SGLT2i) Considering changes in CRT, for instance? Is this difference statistical? Also not clear if non SGLT2i receivet antiVEGF or not. If yes, the comparison between SGLT2i and non SGLt2i should be significant: both with antiVEGF, the only difference SGLT2i.

IT  is important to have the differences between the 2 groups better described, in term of results and final conclusions!

Author Response

This manuscript is a resubmission of an earlier submission. The following is a list of the peer review reports and author responses from that submission.

Round 1

Reviewer 1 Report

The revisions done to the manuscript still do not address the major concerns that were raised previously- 1) very low numbers of patients in the study; 2) missing certain controls; 3) variability in treatment paradigm. Unless more data is added, the results presented in the study will not be conclusive. 

Reviewer 2 Report

In this resubmitted manuscript, the authors addressed all issues raised in the first version by this reviewer. No further, comments. 

Reviewer 3 Report

Diabetic macular oedema represents a retinal complication seen in many diabetic patients. Intravitreal AntiVEGF agents are the prefered regimen so far, with addition of intravitreal triamcinolon. The efficacy is not very high and it is not unusual for a patient to receive 10-20 injections. SGLT2i would be an interesting addition to the treatment, if proved functional.

The main limitation of the study is the low number of patients included (9 plus 4), and the authors are aware of.

SGLT2i regimen should be specified in the MEthod section. It would also be necesary to provide data regarding change in glicemia after SGLT2i innitiation, if available.